# Numerical and Experimental Investigation on a “Tai Chi”-Shaped Planar Passive Micromixer

**DOI:** 10.3390/mi14071414

**Published:** 2023-07-13

**Authors:** Annan Xia, Cheng Shen, Chengfeng Wei, Lingchen Meng, Zhiwen Hu, Luming Zhang, Mengyue Chen, Liang Li, Ning He, Xiuqing Hao

**Affiliations:** 1College of Mechanical and Electrical Engineering, Nanjing University of Aeronautics & Astronautics, Nanjing 210016, China; anxia@nuaa.edu.cn (A.X.); lcmeng@nuaa.edu.cn (L.M.); hzw123@nuaa.edu.cn (Z.H.); zhang-luming@nuaa.edu.cn (L.Z.); mm_chen@nuaa.edu.cn (M.C.); liliang@nuaa.edu.cn (L.L.); drnhe@nuaa.edu.cn (N.H.); 2College of Aerospace Engineering, Nanjing University of Aeronautics & Astronautics, Nanjing 210016, China; cshen@nuaa.edu.cn; 3Beijing Jingdiao Group, Beijing 102308, China; weichengfeng@jingdiao.com

**Keywords:** microfluidic chips, mixing efficiency, pressure drop, simulations, experiments

## Abstract

(1) Background: Microfluidic chips have found extensive applications in multiple fields due to their excellent analytical performance. As an important platform for micro-mixing, the performance of micromixers has a significant impact on analysis accuracy and rate. However, existing micromixers with high mixing efficiency are accompanied by high pressure drop, which is not conducive to the integration of micro-reaction systems; (2) Methods: This paper proposed a novel “Tai Chi”-shaped planar passive micromixer with high efficiency and low pressure drop. The effect of different structural parameters was investigated, and an optimal structure was obtained. Simulations on the proposed micromixer and two other micromixers were carried out while mixing experiments on the proposed micromixer were performed. The experimental and simulation results were compared; (3) Results: The optimized values of the parameters were that the straight channel width *w*, ratio *K* of the outer and inner walls of the circular cavity, width ratio *w*_1_/*w*_2_ of the arc channel, and number *N* of mixing units were 200 μm, 2.9, 1/2, and 6, respectively. Moreover, the excellent performance of the proposed micromixer was verified when compared with the other two micromixers; (4) Conclusions: The mixing efficiency *M* at all *Re* studied was more than 50%, and at most *Re*, the *M* was nearly 100%. Moreover, the pressure drop was less than 18,000 Pa.

## 1. Introduction

Microfluidic chips, which integrate various basic operating units of conventional biochemistry laboratories on a small chip, have been applied to multiple fields, such as analytical chemistry, life sciences, and environmental detection [1,2,3,4]. Benefiting from their high efficiency, low reagent consumption, integration, and portability, micro-analysis has been widely studied and applied [5,6,7]. Thus, micromixers, which play a critical role in the sufficient contact and uniform mixing of reactants, are the focus of researchers [8,9]. In general, micro-scale level mixing depends only on molecular diffusion, where the characteristic scale is generally considered to be within 1 μm–10^3^ μm [10], and the traditional methods of turbulence and stirring are absent, leading to low mixing efficiency.

At present, numerous studies have been performed to enhance the mixing performance. In these studies, micromixers are divided into two categories according to the mixing mechanism, i.e., active and passive mixers. The former requires an external energy source, such as pressure [11,12], electric field [13,14], magnetic field [15,16], and ultrasonic energy [17,18], while the latter relies on the flow channel with a special structure. Compared with active mixers, passive mixers are favored by scholars, since they have the advantages of easy integration, simple operation, high stability, and little effect on the biochemical activity and physicochemical properties of organic fluids [19,20,21,22,23]. 

In passive micromixers, the structure of the flow channel is the key factor affecting mixing efficiency [24]. Early passive micromixers included the T-shape mixer [25] and Y-shape mixer [26] in which the fluids maintained a parallel flow. The mixing efficiency of these mixers was low due to the mixing mechanism of molecular diffusion, which could not meet the increasing requirements. Studies have shown that the enhancement of molecular diffusion and chaotic advection could achieve efficient mixing [22]. Based on this, several studies have been conducted. Veenstra et al. [27] improved the conventional Y-shape by narrowing the main flow channel. The shortened diffusion length enhanced the effect of molecular diffusion. Rahimi et al. [28] reported that an asymmetric inlet, which could cause a violent collision with the incoming fluid, was beneficial to strengthening mixing. Virginie Mengeaud et al. [29] used a finite element model to study the material mixing process of species in a 100-μm-wide zigzag microchannel integrating a “Y” inlet junction. The effects of flow rate and channel geometry on hydrodynamics and mixing efficiency are explained, which laid an excellent foundation for further research. Zhang et al. [30] designed an efficient Y-shape mixer with a zigzag channel in which the fluids were folded constantly, resulting in a chaotic flow. Solehati et al. [31] proposed a micromixer with a wavy-like channel structure based on a linear T-channel micromixer. Their simulation results revealed that the wavy structure could generate a secondary flow within a certain Reynolds number, which would be reversed with the change in the bending direction of the wavy structure. Compared with the flow in the linear T-channel micromixer, the secondary flow disturbed the fluid and promoted the mixing. The split and recombination (SAR) mechanism, which increases the contact area of the mixed fluids, is a new idea for mixing enhancement. The mixing efficiency of a micromixer with 16 channels can reached 95% in 15 ms [32]. Ansari et al. [33] produced a micromixer with an asymmetric circular channel and investigated the influence of the width ratio of the channel on mixing efficiency. Results showed that the differences in the fluid velocity in the channel with different widths resulted in imbalance collision, altering the direction of the fluid in the narrow channel, and enhancing the mixing. In addition, when the width ratio was 2, the mixing efficiency was the best. Channels with obstacles have also been proven to achieve homogenous mixing. Stroock et al. [34] proposed a micromixer with periodic staggered herringbone. The fluids were stretched and folded repeatedly, forming a vortex. Wangikar et al. [35] compared the mixing performance of serpentine micromixers with and without semicircular obstacles. Their results indicated that the mixing efficiency of the former was better, and the larger the radius of the obstacles, the better the efficiency. However, the latter exhibited a smaller pressure drop. Wang et al. [36] also drew the same conclusion by investigating a micromixer with triangular obstacles. In addition, studies on injection mixers [37], droplet mixers [38], and 3D helical mixers [39] have also been reported. These works demonstrated that SAR structures, curved channels, and obstacles can significantly improve the performance of micromixers. Nevertheless, problems, such as high pressure drop and difficult-to-process complex structure, restrict the development of micromixers and introduce new requirements.

In this paper, a novel “Tai Chi” shaped passive micromixer with high efficiency and low pressure drop was proposed to respond to the problems concerning fluid mixing. The influence of the structural parameters including the straight channel width, the ratio of the outer and inner walls of the circular cavity, the width ratio of the arc channel, and the number of mixing units on mixing efficiency and pressure drop were investigated to determine the optimal structure. Subsequently, the performance of the optimized micromixer was verified by comparison with a reflux micromixer and a herringbone groove micromixer. Finally, mixing experiments were performed using the proposed micromixer, and the experimental and simulation results were compared.

## 2. Materials and Methods

### 2.1. Micromixer Model

The geometric structure of the proposed passive micromixer is illustrated in Figure 1a. Two streams of fluid enter from the T-shape inlets and gather into the main channel where the initial contact occurs. Subsequently, the co-fluid flows through several mixing units, which contain a long corner channel, a circular cavity, an arc channel, and a short corner channel. The corner channel can induce vortex generation while the circular cavity and the arc channel can generate secondary flow. Moreover, an unbalanced collision will take place at the exit of the arc channel due to its variable width ratio. The collision and the vortex will enhance the disturbance between the flow layers and result in better mixing efficiency (*M*).

The length and width of the inlet and outlet channels are equal, which are 6 mm and 0.3 mm, respectively. In between, there is the main channel with a length of 0.5 mm and a width of 0.3 mm, and several mixing units whose length is 1630 μm. The structural parameters of the mixing unit, which acts as the main mixing region, have a key impact on mixing performance. In this paper, the width (*w*) of the straight channel, which refers to the channel perpendicular to the axis of the main channel in the corner channel, the ratio (*K = r*_1_/*r*_2_) of the outer and inner walls of the circular cavity, the width ratio (*w*_1_/*w*_2_) of the arc channel, and the number (*N*) of mixing units are taken into consideration.

### 2.2. Governing Equations

Considering that the equations of mass, momentum, and energy conservation are all obtained based on the continuity hypothesis [40,41], the study on the microfluid must clarify whether the continuity hypothesis is still applicable in such a case. The Knudsen number *Kn*, which is defined as the ratio of the mean molecule free walk to a characteristic scale, is generally used to determine it. In this work, the value of the *Kn* is much lower than 10^−3^, satisfying the continuity hypothesis. Therefore, the continuity, Navier–Stokes, and convection-diffusion equations can be solved. The corresponding equations are given below:(1)$$\nabla \cdot (\rho \vec{u})=0,$$
(2)$$\frac{\partial V}{\partial t}+V\cdot \nabla V=-\frac{1}{\rho }\nabla P+\mu \nabla^{2}V,$$
(3)$$D\nabla^{2}c=\vec{u}\cdot \nabla c,$$
where *ρ* is the density, *u* is the velocity, *P* is the pressure, *μ* is the kinematic viscosity, *c* is the concentration, and *D* is the diffusion coefficient.

The mixing degree and energy loss of the fluid in the micromixer are chosen to assess its performance. The mixing degree is defined by the criteria of the concentration variance index [42], while the energy loss is evaluated by measuring the pressure drop (Δ*P*) across the mixing system. Based on the concentration variance index, the mixing efficiency (*M*) can be calculated as:(4)$$\sigma =\sqrt{\frac{1}{N}{\sum_{i=1}^{N}\left(c_{i}-c_{\max}\right)}^{2}},$$
(5)$$M=1-\sqrt{\frac{\sigma^{2}}{\sigma_{\max}^{2}}},$$
where *σ* is the standard deviation of the concentration at one cross-section, *N* is the number of sampling points, *c_i_* is the concentration of sampling point *i*, *c*_max_ is the maximum concentration of all sampling points. According to this equation, *M* = 1 indicates thorough mixing, and *M* = 0 no mixing.

### 2.3. Boundary Conditions

The numerical analysis of the proposed micromixer was performed using COMSOL Multiphysics 5.4. The mixing medium used was deionized water with a density and kinematic viscosity of 1 × 10^3^ kg/m^3^ and 9.7 × 10^−4^ Pa·s, respectively. The modules of laminar flow and transport of diluted species were selected for the numerical simulation. Boundary conditions of velocity inlet and pressure outlet were employed. The same velocities were set and the pressure at the outlet was set as 0. The concentrations of the two inlets were 0 and 1 mol/m^3^, and the no-slip condition was applied to the channel walls. The fluids with different concentrations were marked blue and red to facilitate the visualization of mixing. The mixing of fluids was investigated under different Reynolds numbers *Re*, whose expression is: (6)Re=ρuLμ,
where *ρ* is the density, *u* is the velocity, *L* is the characteristic scale and *μ* is the kinematic viscosity. The correspondence between *Re* and inlet velocity is given in Table 1.

### 2.4. Mesh Independence Test

Mesh generation is an important part of numerical simulations. With the increase of domain elements, the accuracy of the results would certainly be improved. However, in this regard, the calculation time would be therefore further prolonged. Simulations were performed on the same model with different domain elements to obtain a balance between result accuracy and computational efficiency. The numbers of elements were selected as 62.65 × 10^4^, 123.36 × 10^4^, 210.82 × 10^4^, 291.63 × 10^4^, and 326.51 × 10^4^. Due to the local complex structure of the proposed micromixer, a hybrid mesh is employed, i.e., a structured mesh was used for the regular parts, such as the inlet and main channels, while an unstructured mesh was used for the complex parts, such as the mixing units. The *M* and pressure drop under different domain elements are presented in Figure 2. It can be observed that the number of elements had almost no effect on pressure drop, while *M* tended to sharply decrease with the increase of elements number. The *M* decreased by 2.64%, 1.12%, 0.48%, and 0.09%, respectively. Therefore, to improve the numerical simulation efficiency while ensuring high accuracy, the domain elements of 210.82 × 10^4^ were selected.

### 2.5. Preparation of Micromixer

In this paper, the mold replication method is used to process and manufacture the micromixer. Precision glass molding technology is the most efficient method of manufacturing micro/nanostructured glass components [43]. C45w mold steel material is selected as the material of the micromixer mold, and Polymethyl methacrylate (PMMA) is selected as the material of the micromixer chip. The method of machining micromixer mold by micro-milling is presented. Micro-milling technology has the advantages of high forming precision, good reliability, high efficiency, wide material applicability, and can process complex three-dimensional surface structures, etc., which fully has the ability of micromixer mold processing and manufacturing [44]. First, the micromixer mold was processed in the ultra-precision micro-milling processing center, and the size of the workpiece is 25 × 25 × 5 mm. To ensure processing accuracy and improve processing efficiency, the milling cutter with a diameter of 0.5 mm was used to roughen the micro mold. After removing a large amount of material, the milling cutter with a diameter of 0.2 mm was used for finishing to obtain the profile topography of the micro die. The machining parameters are shown in Table 2. The bottom surface roughness, side-wall roughness, and side-wall verticality of the machined microgroove under this parameter were 279 nm, 348 nm, and 91.8°. After the micro mold processing was completed, the liquid PMMA was poured into the micro mold, and the substrate of the micromixer channel was obtained by demolding. Then, the obtained microchannel substrate was sliced to obtain the substrate of the required size, and it was assembled with the cover plate with liquid inlet and outlet and the same size by hot-pressing bonding process to obtain the final micromixer chip, as shown in Figure 3.

### 2.6. Experiment on Mixing Performance of Micromixer

The mixing experiments were conducted on a self-built experimental platform using the micromixer with an optimized structure (Figure 4). The selected mixing media were deionized water and inked deionized water. During the experiment, the two streams of fluid can be injected into the micromixer at a certain velocity through a micro-syringe pump. A charge-coupled device (CCD) camera placed above the mixer was used to observe the fluid flow in real time, while a differential pressure transmitter connected the inlet and outlet was employed to measure the pressure drop of the entire mixing system. The *M* can be obtained by performing grayscale processing on the images captured by the CCD camera, according to the following equations:(7)$$\sigma =\sqrt{\frac{1}{n}\sum_{i=1}^{n}{\left(I_{i}-\bar{I}\right)}^{2}},$$
(8)$$\bar{I}=\frac{1}{n}\sum_{i=1}^{n}I_{i},$$
(9)$$M=\frac{\sigma_{\max}-\sigma_{s}}{\sigma_{\min}-\sigma_{\min}},,$$
where *I_i_* is the gray value of sampling point *i*, *n* is the number of sampling points, $\bar{I}$ is the average gray value of all sampling points, and *σ*_max_ and *σ*_min_ are the standard deviations of the gray values in the sampling area when the deionized water and the inked deionized water are completely unmixed and completely mixed, respectively. In addition, considering that the driving parameter of the syringe pump is flow rate *q*, the corresponding relationship between *q* and *Re* should be calculated in advance. The flow range corresponding to *Re* ranging from 0.1 to 80 is given in Table 3.

## 3. Results and Discussion

### 3.1. Effect of the Width of the Straight Channel on Mixing Performance

On the one hand, when the fluid flows from the long corner channel to the circular cavity, the expansive structure can induce a vortex, on the other hand, the fluid, flowing out of the arc channel, can produce an unbalanced collision in the short corner channel. The width *w* has a significant influence on the magnitude of the vortex and collision. Micromixers with a straight channel width *w* of 100 μm, 200 μm, and 300 μm were investigated. When *Re* ranges from 0.1 to 80, the *M* and pressure drop of the three mixers are displayed in Figure 5. It is found that the relationship between *M*, Δ*P*, and *Re* are similar in all micromixers, i.e., with the increase of *Re*, *M* tends to decrease and then increase, while Δ*P* exhibits an increasing trend. As shown in Figure 6, at a low *Re* of 0.1, the velocity of the fluid was very low and did not exceed 0.0035 m/s, so that the fluid has more time to fully contact the channel and make it mixed evenly. Therefore, the *M* values of the three mixers are all close to 100%. As *Re* changes from 0.1 to 1, the mixing is still dominated by molecular diffusion; however, the increasing velocity reduces the residence time, resulting in the decrease of *M* (Figure 5a). Subsequently, the progressively higher velocity gradually induces collision and vortex generation and improves the mixing degree. Nevertheless, when *Re* increases to a certain value, the mixing reaches saturation, and *M* approaches 1. In addition, the *M* of the micromixer with *w* = 100 μm is the highest at Re of 1–20, which can be explained by the streamlines illustrated in Figure 6. In the case of low *Re*, the smaller the *w*, the stronger the molecular diffusion, which allows for more thorough mixing in the channel connected to the arc channel when fluid flows from the arc channel to the short corner channel. It can be seen from Figure 6a that in the micromixer with *w* = 100 μm, the fluid flow rate at the A-A section is greater than that in the mixer with larger *w*. The streamlines of the two fluids are crossed (*Re* = 1), and some vortices appeared at the edge of the channel, which promotes the mixing efficiency of the fluid. However, in micromixers with larger *w*, the streamlines are non-interfering. When *Re* is substantially high, the smaller *w* can lead to more disordered streamlines and stronger vortices (*Re* = 10), which indicates that severe disturbance occurs, and the flow layer is destroyed. At this time, part of the fluid changes into a spiral forward, which is speculated to produce a certain degree of convective diffusion, increasing the contact area between the fluids, and thus greatly improving the mixing efficiency. At a high *Re* of 40, the smaller *w* is, the faster the flow velocity is, the more disordered the streamlines are, the more vortices are generated and the larger the vortex range is. This phenomenon proves that the smaller *w* is conducive to improving mixing efficiency. As for Δ*P*, it depends on linear loss and local loss. Linear loss is related to *Re*, while local loss is determined by *Re* and the structure where it occurs. In general, the larger the *Re* and the more complex the local structure, the greater the Δ*P*. Since the strongest collision and vortex are observed in the micromixer with *w* = 100 μm, its Δ*P* is also the highest, and especially at higher *Re*, the increase of Δ*P* is more significant. Under comprehensive consideration, the optimal value of *w* was selected as 200 μm. 

### 3.2. Effect of the Ratio of the Outer (r_1_) and Inner (r_2_) Walls on Mixing Performance

The structure of the circular cavity can induce secondary flow in the micromixer, strengthening the fluid collision and significantly improving the mixing efficiency. Micromixers with a *K* of 1.63, 1.95, 2.9, 3.85, and 6.7 were studied, and the corresponding *r*_2_ were 900 μm, 600 μm, 300 μm, 200 μm, and 100 μm, respectively. The *M* and pressure drop of the micromixers with different *K* under *Re* ranging between 0.1 and 80 are presented in Figure 7. At too low or too high *Re*, high-intensity molecular diffusion or chaotic convection is conducive to enhancing the mixing, and *M* is close to 100%. It can be seen that when *Re* ranges from 1 to 40, the larger the *K*, the greater the increase rate of *M.* As shown in Figure 7a, the *M* of the micromixer with *K* = 6.7 is the lowest at *Re* = 1 and the largest at *Re* = 40. When *Re* = 1, the mixing depends mainly on molecular diffusion, there is no vortex at the A-A section (Figure 8), and the length of the micromixer with the smaller *K* is longer, which promotes the mixing efficiency. With the increase of *Re*, the flow at the A-A section tends to generate a vortex and the streamlines become disordered (*Re* = 10). The larger *K* indicates that the difference in the velocity at the two walls is higher, which is more conducive to vortex generation and mixing efficiency improvement. Moreover, due to the large curvature of the inner wall and the obvious adhesion effect of the wall, it is easier to produce separation vortices when the fluid flows through, resulting in a greatly improved the *M* of the micromixer with large *K* values. In addition, the longer channel length in the micromixer with a smaller *K* induces a higher pressure drop. Considering the values of *M* and Δ*P* obtained under all *Re* studied, it was concluded that the optimal value of *K* is 2.9, which corresponds to an *r*_2_ of 300 μm.

### 3.3. Effect of the Width Ratio (w_1_/w_2_) of the Arc Channel on Mixing Performance

The arc channel of the proposed micromixer refers to the study of Ansari [32], i.e., the width ratio is selected as 2, which can lead to an unbalanced collision between the two fluids during recombination due to differences in the flow velocity. Considering the asymmetrical structure of the proposed mixer, the micromixers with *w*_1_/*w*_2_ = 1/2 and *w*_1_/*w*_2_ = 2/1 were investigated. Figure 9 demonstrates the mixing efficiency and pressure drop of the two micromixers at *Re* of 0.1–80. When *Re* is too high (*Re* > 40) or too low (*Re* < 1), the *w*_1_/*w*_2_ has little effect on efficiency. This is because the mixing depends on molecular diffusion at low *Re*, and the channel length plays a key role in the mixing degree, while the effect of the structure is negligible. At high *Re*, a high-intensity vortex and secondary flow are generated, resulting in saturated mixing. At medium *Re*, the weaker molecular diffusion and chaotic convection make the width ratio have a significant influence on *M*, and as it can be seen in Figure 9a, the micromixer with *w*_1_/*w*_2_ = 1/2 has obvious advantages. Moreover, by comparing Figure 9a,b, it can be found that the Δ*P* of the micromixer with *w*_1_/*w*_2_ = 1/2 at low *Re* is almost the same as that of the one with *w*_1_/*w*_2_ = 2/1, and Δ*P* at high *Re* is even lower. This indicates that the performance of the micromixer with *w*_1_/*w*_2_ = 1/2 is much better.

To explore the significant advantages of the micromixer with *w*_1_/*w*_2_ = 1/2, the fluid flow at the B-B section was investigated (Figure 10). It can be seen that under all *Re* studied, a pair of vortices appears in the micromixer with *w*_1_/*w*_2_ = 1/2, while only gradually disordered streamlines occur in the one with *w*_1_/*w*_2_ = 2/1. This is because, in the micromixer with *w*_1_/*w*_2_ = 1/2, the flow rate difference between the center and the edge of the channel is large, the small flow rate at the edge leads to low pressure, and the large flow rate in the central region leads to high pressure. At this time, the fluid passes through a curve with a large curvature ratio, and the pressure gradient and centrifugal force make the fluid return flow, resulting in the appearance of secondary flow. In the micromixer with *w*_1_/*w*_2_ = 2, although the pressure difference between the center and the edge of the channel is also generated, the curvature ratio of the curve flowing through it is small, so no secondary flow is generated. The secondary flow at the B-B section intensifies the disturbance of the flow layers, which significantly improves the mixing efficiency. In addition, the pressure loss is not found to be much different, since the structure is similar. In general, the micromixer with *w*_1_/*w*_2_ = 1/2 is preferred.

### 3.4. Effect of the Number of Mixing Units on Mixing Performance

The *M* is closely related to the length of the mixing channel. The longer the channel, the longer the residence time, which leads to a higher *M*. However, a longer channel would increase the mixing time and would not be suitable for applications requiring rapid mixing. Moreover, after a certain length, the effect of channel length on *M* is insignificant. The number of mixing units should be reasonably selected to achieve a balance between *M* and mixing time. The *M* and pressure drop of micromixers with 4, 5, 6, 7, and 8 mixing units are displayed in Figure 10. When *Re* > 0.1 or *Re* < 40, *M* increases with the increase of *N*, but the increase rate tends to be progressively smaller. According to Figure 11a, at high *Re*, the *M* of the micromixers with different *N* is almost the same. This means that the mixing has reached saturation. Moreover, *N* and Δ*P* are positively correlated, i.e., the larger the *N*, the higher the Δ*P*, especially at higher *Re*. After comprehensive consideration, an *N* of 6 was selected to guarantee high efficiency and low pressure drop.

### 3.5. Comparison with Other Micromixers

To verify its superiority of performance, simulations were carried out on the proposed micromixer and the other two micromixers. The *w*, *K*, *w*_1_/*w*_2_, and *N* of the proposed micromixer were 200 μm, 2.9, 1/2, and 6, respectively. The other two micromixers were a reflux micromixer and a herringbone groove micromixer (Figure 12). For both, *N* = 6 was also selected. In the modeling process, the size of the inlet and outlet section, the length of the inlet and outlet channel, and the length of the single mixing unit of the three micromixers are equal. Considering that the *M* of the proposed micromixer is nearly 100% at very low and high *Re*, the range of *Re* investigated was between 1 and 40. The *M* and pressure drop of the three micromixers are shown in Figure 13. As it can be seen in Figure 13a, the *M* of the proposed micromixer is lower than that of the herringbone groove micromixer only when *Re* = 1 and *Re* = 5, and larger under all other *Re*. Moreover, the Δ*P* of the herringbone groove micromixer is the largest over the entire *Re* range studied. As for the reflux micromixer, although its Δ*P* is the lowest, its *M* is significantly lower than that of the other two micromixers. Consequently, considering the *M* and pressure drop, the proposed micromixer exhibits an excellent mixing performance.

### 3.6. Experiment Analysis Using the Proposed Micromixer

The fluid flow in the micromixer is shown in Figure 14 under different values of *Re* ranging from 0.1 to 80. It can be observed that the fluid flow is relatively stable, and there is an obvious molecular diffusion phenomenon. More specifically, the ink diffuses gradually into the deionized water in the second, third, and fourth mixing units, resulting in an increasingly blurred interface. With the increase of *Re* (*Re* = 1 and *Re* = 5), a clear interface can be observed in the entire flow channel, which is due to the weaker molecular diffusion effect. Only the mixing interface of deionized water and ink exhibits a little molecular diffusion. When *Re* = 10, the disturbance is generated. Compared with the flow pattern at the outlet when *Re* = 5, it can be found that the mixing degree is significantly enhanced, and the appearance of the mixed medium is relatively uniform. A vortex appears in the circular cavity and the color difference between the two fluids becomes less apparent when *Re* is increased to 20. In addition, as *Re* continues to increase, the intensity of the vortex has become gradually stronger. It can be seen that when *Re* = 40 and *Re* = 80, the mixing interface after several mixing units becomes blurred, and in the sixth mixing unit at *Re* = 40 and the fifth mixing unit at *Re* = 80, the color of the fluid is highly uniform, which indicates that the mixing has reached saturation.

Figure 15 and Figure 16 compare the experimental and numerical simulation results. Typical *Re* (*Re* = 0.1, 1, 10, and 40) was selected to analyze the fluid mixing, and for each *Re*, the flow pattern in the third, fourth, and fifth mixing units were obtained. It can be observed that when *Re* = 0.1, the fluid flow in the experimental and simulation results is similar with no obvious unstable flow. The concentration contours show that the concentration in the third, fourth, and fifth mixing units is close to 0.5, which indicates even mixing and is in accord with the experimental results. When *Re* = 1, both experimental and simulation results display a poor mixing degree. At *Re* = 10, the streamlines of the circular cavity near the inner circular wall begin to be turbulent, while the same phenomenon appears in the experimental results. When *Re* = 40, the disturbance of the fluid flow is similar to that of the streamlines. Therefore, it can be concluded that the experimental results are in good agreement with the simulation ones, which suggests that numerical simulations can provide certain guidance for the experiments. Figure 16 demonstrates and compares the experimental and simulation results of the *M* and pressure drop. Under most *Re*, the *M* obtained experimentally is lower than that obtained in the simulation, while the pressure-drop results presented an opposite trend. The reason for this is that in the numerical simulation, fluid mixing is defined as an ideal situation, while in the experimental process, the accuracy of the instruments, the fabrication of the micromixer, the boundary layer effect, and the possible errors in gray processing may have a certain effect on the final experiment results. In general, the difference between experiment and simulation results is small, which can be considered to be in good agreement.

## 4. Conclusions

Aiming at the problems of low efficiency and high pressure drop, a novel Tai Chi passive micromixer was proposed. The conclusions are as follows:(1)The mixing state changes from molecular diffusion to chaotic convection with the increase of *Re*. At very low (*Re* < 1) and high (*Re* > 40) *Re*, the *M* is almost 100%. Nevertheless, in the case of medium *Re*, the intensity of the interaction of the two mechanisms is not high, leading to low *M*. In addition, the larger the *Re*, the higher the velocity, and the higher the pressure loss, especially at larger *Re*, the higher the increase of pressure drop.(2)Furthermore, the smaller the *w*, the larger the *M* and pressure drop. The effect of *K* on mixing performance is mainly reflected under medium *Re*, where the larger *K* results in a larger increase in the *M* and a smaller pressure drop. The *w*_1_/*w*_2_ has a significant effect on mixing performance, and the micromixer with *w*_1_/*w*_2_ = 1/2 was preferred. With the increase of *N*, the *M* and pressure drop tend to increase. Considering the *M* and pressure drop under different structural parameters at different *Re* ranging from 0.1 to 80, the structural parameters were optimized to be *w* = 200 μm, K = 2.9, *w*_1_/*w*_2_ = 1/2, and *N* = 6.(3)The experimental results demonstrated that the *M* at all *Re* studied was more than 50%, and at most *Re*, the *M* was nearly 100%. Moreover, the pressure drop was less than 18,000 Pa. The experimental and simulation results were in good agreement, which shows that numerical simulations have certain accuracy and can provide guidance for subsequent experimental research.

## Figures and Tables

**Figure 1 micromachines-14-01414-f001:**
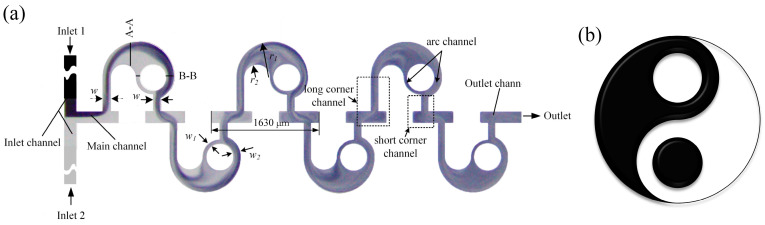
(**a**) Schematic diagram of the Tai Chi-shaped micromixer and (**b**) Chinese Tai Chi Pattern.

**Figure 2 micromachines-14-01414-f002:**
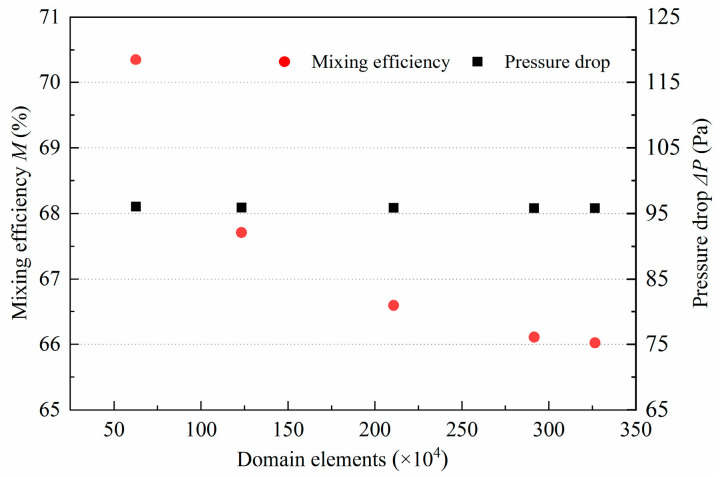
Mesh independence test.

**Figure 3 micromachines-14-01414-f003:**
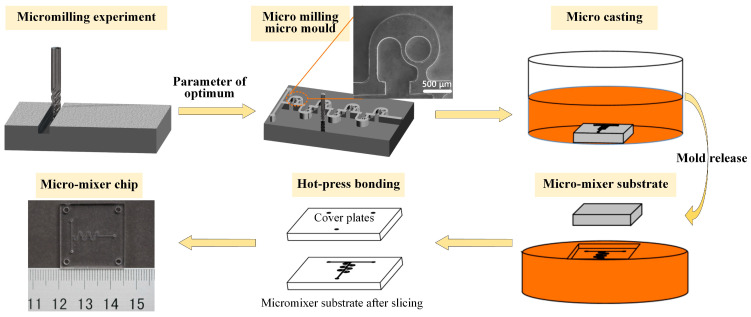
The manufacturing process of micromixer.

**Figure 4 micromachines-14-01414-f004:**
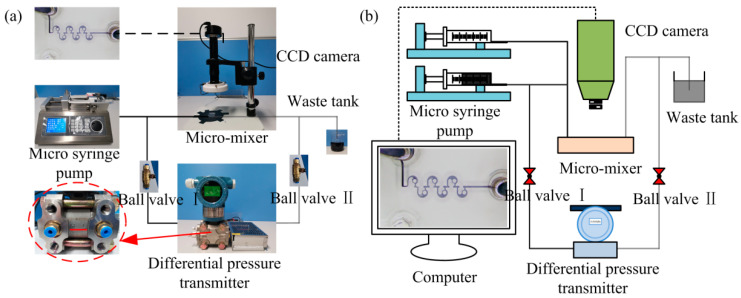
(**a**) Images and (**b**) schematic diagram of the self-built experimental platform.

**Figure 5 micromachines-14-01414-f005:**
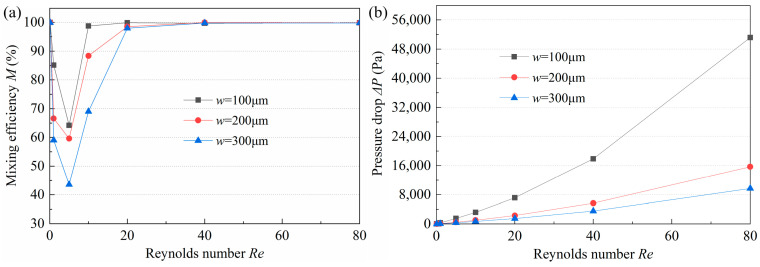
Influence of *w* on (**a**) mixing efficiency and (**b**) pressure drop.

**Figure 6 micromachines-14-01414-f006:**
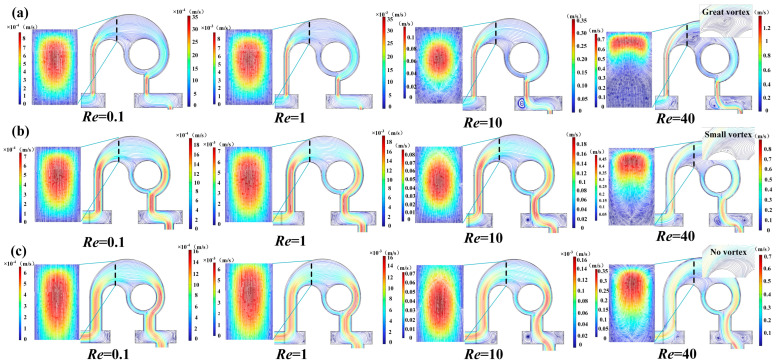
Simulation results of flow velocity in the micromixers with (**a**) *w* = 100 μm, (**b**) *w* = 200 μm, (**c**) *w* = 300 μm at different *Re* numbers.

**Figure 7 micromachines-14-01414-f007:**
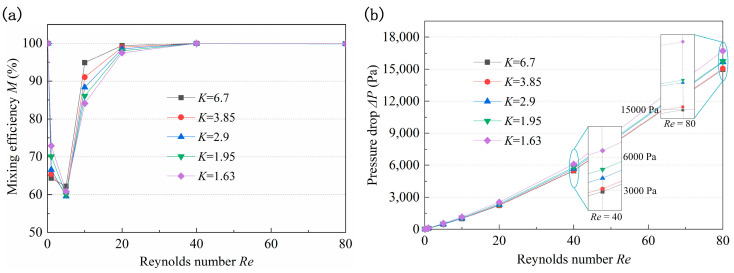
Influence of *K* on (**a**) mixing efficiency and (**b**) pressure drop.

**Figure 8 micromachines-14-01414-f008:**
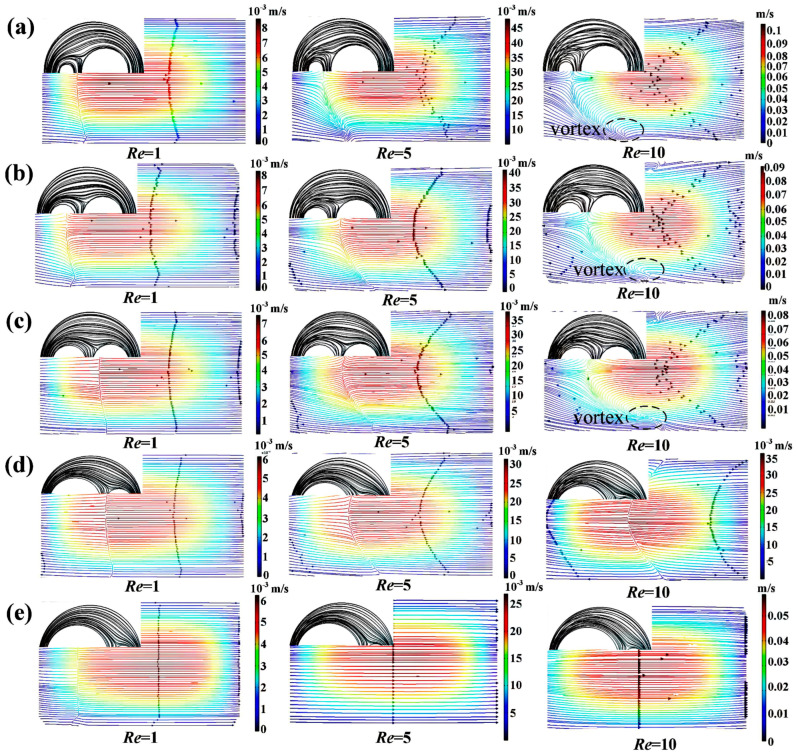
Simulation results of streamlines of velocity at the A-A section (Figure 1) and in the circular cavity of the micromixers with (**a**) *K* = 6.7, (**b**) *K* = 3.85, (**c**) *K* = 2.9, (**d**) *K* = 1.95, and (**e**) *K* = 1.63 at typical *Re*.

**Figure 9 micromachines-14-01414-f009:**
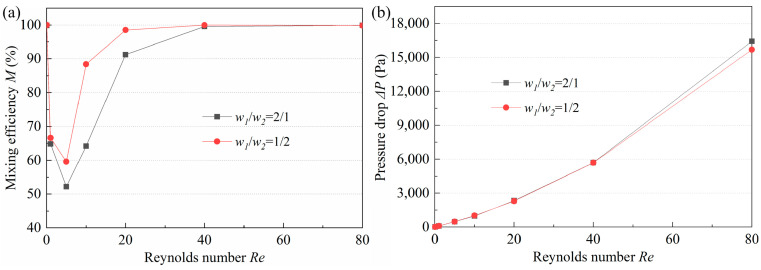
Influence of *w*_1_/*w*_2_ on (**a**) mixing efficiency and (**b**) pressure drop.

**Figure 10 micromachines-14-01414-f010:**
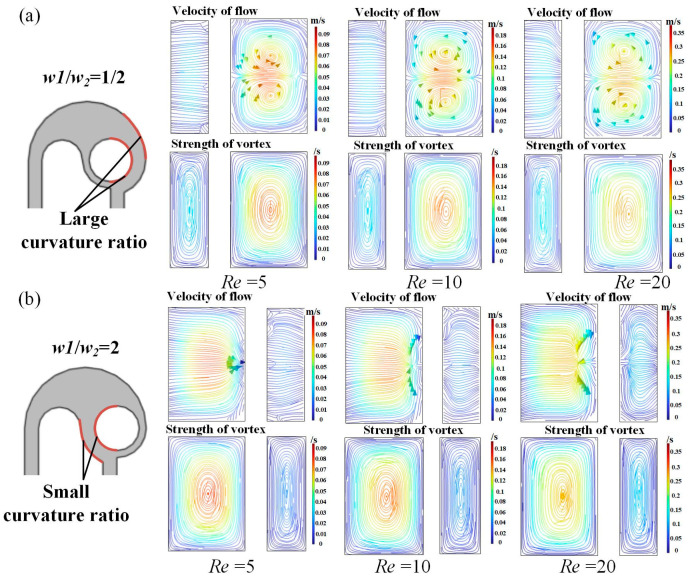
Simulation results of flow velocity and vortex strength at the B-B section of the micromixers with (**a**) *w*_1_/*w*_2_ = 1/2 and (**b**) *w*_1_/*w*_2_ = 2/1 at typical *Re*.

**Figure 11 micromachines-14-01414-f011:**
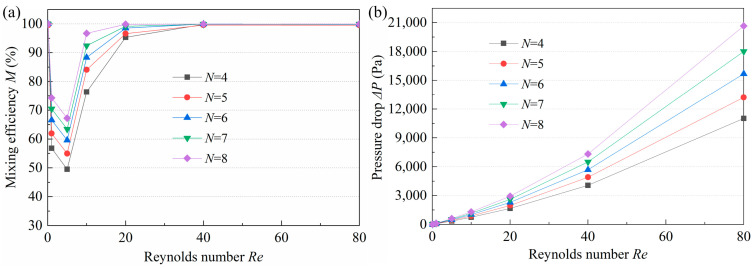
Influence of *N* on (**a**) mixing efficiency and (**b**) pressure drop.

**Figure 12 micromachines-14-01414-f012:**
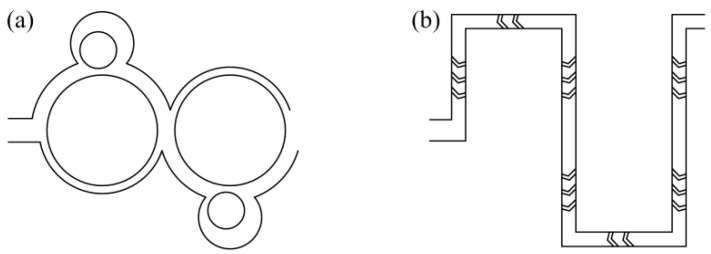
Schematics of the (**a**) reflux and (**b**) herringbone groove micromixer structures.

**Figure 13 micromachines-14-01414-f013:**
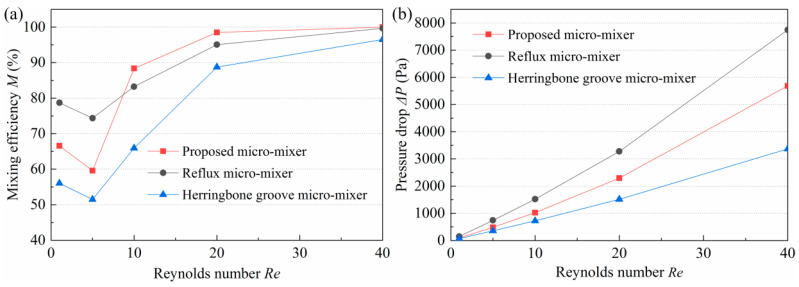
Comparison of the (**a**) mixing efficiency and (**b**) pressure drop of the proposed micromixer and the other two micromixers.

**Figure 14 micromachines-14-01414-f014:**
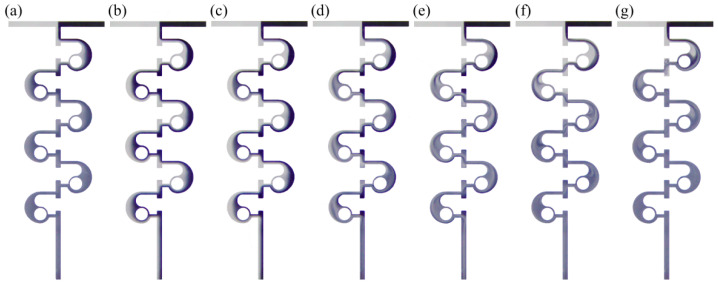
Experimental results of fluid flow state in a micromixer at different *Re* numbers: (**a**) *Re* = 0.1; (**b**) *Re* = 1; (**c**) *Re* = 5; (**d**) *Re* = 10; (**e**) *Re* = 20; (**f**) *Re* = 40; (**g**) *Re* = 80.

**Figure 15 micromachines-14-01414-f015:**
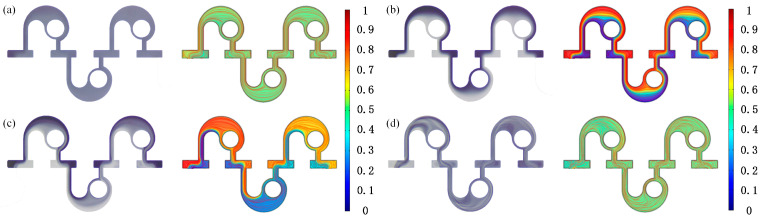
Comparison between the fluid flow obtained by experimental and simulation at different *Re* numbers: (**a**) *Re* = 0.1; (**b**) *Re* = 1; (**c**) *Re* = 10; (**d**) *Re* = 40.

**Figure 16 micromachines-14-01414-f016:**
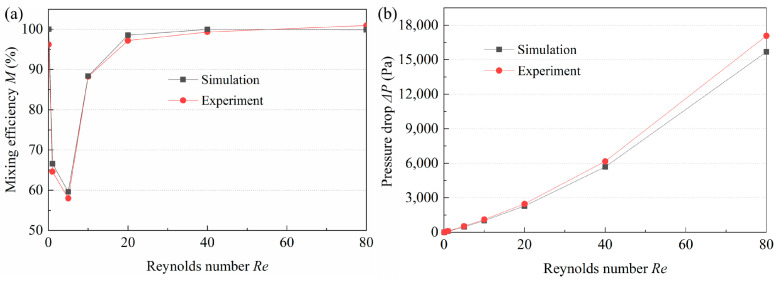
Comparison of the (**a**) mixing efficiency and (**b**) pressure drop between the experimental and simulation results.

**Table 1 micromachines-14-01414-t001:** Correspondence between Reynolds number *Re* and inlet velocity *u*.

Reynolds number *Re*	0.1	1	5	10	20	40	80
Inlet velocity *u* (×10^−2^ m/s)	0.032	0.32	1.62	3.23	6.47	12.93	25.87

**Table 2 micromachines-14-01414-t002:** Milling parameters of micromixer mold.

Cutter Diameter (mm)	Spindle Speed (r·min^−1^)	Feed per Tooth (μm/z)	The Axial Cutting Depth (μm)	The Radial Cutting Depth (μm)
0.2	40,000	2.5	8	120
0.5	30,000	5.333	15	200

**Table 3 micromachines-14-01414-t003:** Correspondence between Reynolds number *Re* and flow rate *q*.

Reynolds number *Re*	0.1	1	5	10	20	40	80
Flow rate *q* (μL/min)	1.746	17.46	87.3	174.6	349.2	698.4	1396.8

## Data Availability

Not applicable.

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
