# Peer review of "Numerical and Experimental Investigation on a “Tai Chi”-Shaped Planar Passive Micromixer"

_micromachines, 2023, doi:10.3390/mi14071414_

Round 1
Reviewer 1 Report
1. Lines 98-112 on page 3 are irrelevant.
2. The clarity of all charts needs to be improved.
3. In addition to the streamlines, velocity contours need to be added to further explain the hydrodynamic effects in the microchannel.
4. The use of only two-dimensional models in numerical simulation can ignore the secondary flow in the cross section of the channel, which has been found to play a very important role in many micromixer-related studies.
5. It is not convincing to explain the effect of width on mixing efficiency only on the basis of the crossing of streamlines at low Re. In addition, at high Re (e.g. Re=40), there is no significant difference in the size of vortices and the degree of streamline confusion in the diagrams of the three channels. It is suggested to supplement the velocity field in the horizontal plane and cross section of the channel for further explanation.
6. On page 8, line 270, the M and pressure drop of micromixers with different K should be shown in Figure 7 instead of Figure 6.
7. In Figure 7b, only the variations of pressure drop under the three K-value conditions of 1.63, 1.95 and 3.85 can be seen, and the difference is not significant, so it is suggested to supplement the local enlarged figure at the key trend.
8. As shown in Figure 8, the differences of streamlines under different flow conditions are not obvious, it is suggested to add labels to distinguish and explain the evolution trend of streamlines.
9. It is very limited to simply describe the mechanism of the influence of each variable on the mixing efficiency with only molecular diffusion and chaotic convection. The fluorescence particle tracer experiment diagram should be added to illustrate the real flow patterns in the channel.
10. In addition to the observed streamlines, quantified parameters are needed to further describe the vortex intensity in the channel (e.g., vortex area, velocity distribution, etc.)
11. In lines 304-307, why do a pair of vortices appear at w1/w2=1/2 while only gradually disordered streamlines appear at w1/w2=2/1? The differences of streamlines under different conditions need further explanation.
12. Why choose the reflux micromixer and the herringbone micromixer in the manuscript for comparison? In fact, more micromixers, such as inertial micromixers and split-recombination micromixers, have good mixing performance and have been widely studied. In addition, the comparison under different structural and dimensional parameters lacks practical significance.
-
Write fluently
Reviewer 2 Report
The manuscript "Numerical and experimental investigation on a ‘Tai Chi’ shaped planar passive micro-mixer" addresses an important problem of microfluidic mixing. It proposes a new device design for a mixer, which minimizes pressure drop and provides efficient mixing.
The research idea is interesting and the topic of the manuscript fits the scope of Micromachines and the topic of the special issue.
Before considering for publication, a revision of the manuscript is required. The following major and minor issues should be addressed:
1) The abstract lists the structure of the manuscript that is not typical for research paper abstracts. It is recommended to revise the abstract is a more coherent and integral way so it summarizes problem statement and paper aim as well as main results and conclusions.
2) Microfluidic mixing is a vibrant area, which is rapidly developing today. The list of references, however, contains mostly 8-10 years old papers and not so many newer manuscripts. The authors are recommended to add more research results published within last 3-5 years to their review of literature and the list of references.
3) Conclusions seem to be too long and hard to read (especially, what is the reason for the first paragraph? Does it introduce conclusion in the Conclusions section?) . The authors are recommended to shorten their conclusion by briefly summarizing the main outcomes of the paper and their significance.
4) It is necessary to check formatting of the manuscript (the font in the last paragraph of Conclusion is different.
5) Line 37 - it is recommended to define the term "macro analysis" more precisely. Some experimental techniques, which are traditionally considered as "macroscopic" rely on microscale or nanoscale effects.
6) Line 60. Was the reference [29] the first citation of zigzag mixing? Such a microfluidic technique was reported much earlier, for example Anal. Chem. 2002, 74, 16, 4279–4286.
7) Lines 98-112. The remaining text from the Micromachies template is present. The authors should carefully revise their manuscript to assure that the revised version has no such imperfections.
8) Fig. 1 - check the figure quality (a higher contrast alternative is recommended - same for other figures). Also, it looks like that after six cycles, there is no mixing at all (same color gradient - that is different in Fig. 14, for example).
9) Line 126 - is the total length 1630 mm? Please comment.
10) Sections 2 and 3 are recommended to combine into a single Materials and method section. This section is recommended to be also shortened and provide brief experimental details about microchip fabrication, numerical analysis and mixing experiments. Continuity hypothesis discussion, governing equations, etc. are more appropriate for the Results and Discussion section.
11) Table 3 - Reynolds numbers are inconsistent; the authors should check them.
12) Line 230 - it is not recommended to use abbreviated terms in headings.
The quality of English is good. Moderate check is recommended.
Round 2
Reviewer 1 Report
I have no suggestion now
Reviewer 2 Report
The authors addressed my questions and concerns in the revised manuscript. It is recommended for publication.
The quality of English is high enough to render the science clear. Minor check is recommended to assure that typos such as that in Line 1 will be corrected during further processing.